# OpenReview forum: "Learning to Align, Aligning to Learn: A Unified Approach for Self-Optimized Alignment"
_ICLR.cc/2026/Conference — ICLR 2026 Conference Withdrawn Submission_

### Official Review · Reviewer_dvMB · 2025-10-28

**Soundness:** 1
**Presentation:** 1
**Contribution:** 1
**Rating:** 0
**Confidence:** 5

**Summary:**

This paper introduces **GRAO (Group Relative Alignment Optimization)** — a framework that unifies **Supervised Fine-Tuning (SFT)** and **Reinforcement Learning (RL)**–based alignment. It combines a **Group Relative Optimization (GRPO)**–style reward objective with an **SFT loss** to balance imitation and exploration.

The proposed loss (Eq. 3) integrates normalized advantages, group-relative weighting, and a reference term, generalizing prior methods like **DPO** and **GRPO**.
Theoretical analysis claims **convergence guarantees** under standard assumptions, and experiments on *helpful* and *harmless* alignment benchmarks show improvements over SFT, PPO, DPO, and GRPO.

**Claims:** GRAO unifies SFT and RLHF under one framework, improves **sample efficiency**, provides **theoretical convergence guarantees**, and claims enhanced **reasoning ability**.

**Strengths:**

No Strengths

**Weaknesses:**

#### **1. Incomplete and Ambiguous Objective Formulation**
- **Equation (2)** (Line 215) defines an expectation but omits the inner function $f(X)$, making the expression incomplete.
- **Equation (3)** (Line 219) references advantage terms $\hat{A}_{o_i}$ and $\hat{A}_y$ without defining the reward function $R(o_i, y)$ (Line 250).
- Several variables remain undefined throughout the paper:
  - $o_i$ in Eq. (5)
  - $o_{\text{pre}, i}$, $o_{\text{post}, i}$ in Eq. (6)
- These omissions make it difficult interpret the proposed optimization algorithm.

#### **2. Lack of Optimization Method Description**
- Section 3.2 presents the GRAO objective but **does not specify how it is optimized**.
- The link between the theoretical convergence proof and the actual optimization implementation is unclear.

#### **3. Metrics and Evaluation Ambiguity**
- The proposed metrics (**RAS** and **NAG**) in Section 4.1 are insufficiently defined.
- **RAS** relies on comparing model and reference rewards using  $R(o_{i}, y)$, and there is no clear definition of this Reward Computation
- Similarly, **NAG** depends on $R(o_{\text{pre}, i}, y_{\text{ref}, i})$ and $R(o_{\text{post}, i}, y_{\text{ref}, i})$, yet these reward computations are **not defined anywhere**.

#### **4. Conceptual Vagueness and Poor Clarity**
- The phrase **“unified alignment problem”** (Line 041) is introduced without definition.
- **“The recently released model”** (Line 044) is ambiguous — unclear which model is being referenced.
- Statements such as *“it has the ability of quite complex reasoning tasks”* (Line 048) are grammatically incorrect and scientifically vague.
- The conceptual novelty over **GRPO** and **SFT** appears incremental.

#### **6. Limited Evaluation Scope**
- Despite strong claims about “complex reasoning” improvement, **no reasoning datasets** are used — only *helpful* and *harmless* alignment benchmarks.
- The experiments therefore do not support the claimed advances in reasoning or generalization.

**Questions:**

#### **1. Clarification of the Optimization Technique**
- The paper presents the GRAO objective in Eq. (3), but the optimization procedure remains unclear.
  - Please specify the exact optimization technique used for minimizing this objective.

#### **2. Missing Definition of Reward and Trajectory Variables**
- The core reward term $R(o_i, y)$ is central to computing normalized advantages  $\hat{A}_i = \frac{R(o_i, y) - \mu_r}{\sigma_r},$  yet the paper never defines how $R(o_i, y)$ is obtained.
  - In Eq. (5), the definition of $R(o_i, y_{\text{ref}, i})$ is missing.
  - In Eq. (6), the quantities $o_{\text{pre}, i}$ and $o_{\text{post}, i}$ are introduced without definition — please clarify what these represent (e.g., pre- and post-update trajectories).
  - Similarly, $o_i$ (Eq. 5) is used but never formally defined.

#### **3. Reward Statistics (Mean, Variance, Quartiles)**
  - Please report the **mean**, **variance**, and **quartile ranges** of the raw reward values $R(o_i, y)$ on the evaluation dataset.


#### **4. Evaluation on Reasoning and Mathematical Datasets**
- The paper repeatedly claims that GRAO improves “complex reasoning” capability, but evaluation is restricted to *helpful* and *harmless* alignment benchmarks.
  - To substantiate the reasoning claim, please include evaluations on **reasoning-oriented datasets** such as GSM8K, MATH, or AIME. This  would demonstrate whether GRAO contributes to reasoning ability beyond behavioral alignment.

#### **5. Explanation of Theoretical Guarantees in Terms of Sample Complexity**
- Abstract states that the proposed algorithm offers *convergence guarantees* and *sample efficiency advantages*, but the theoretical formulation (Eq. (4)) only proves gradient-norm convergence under standard assumptions.
  - Could you explicitly connect these theoretical results to the sample-complexity?

---

### Official Review · Reviewer_8Mni · 2025-10-31

**Soundness:** 2
**Presentation:** 2
**Contribution:** 2
**Rating:** 2
**Confidence:** 4

**Summary:**

The paper proposes an unique way to train language models that combines learning from humans (SFT) and learning from feedback (RLHF) into one single process. Instead of the standard process of first training the SFT and initializing that for RL, the paper proposes a joint optimization framework via imitating good human answers, explore its own better answers, and maintain distance to reference policy. Empirical results are promising on helpfulness harmless task.

**Strengths:**

- The approach shows an unique way of comibing SFT and RL with imitiation learning and exploration type losses
- Empirical analysis is promising showing improvement over baselines which shows faster convergence over baselines
- Works well on both normal (dense) and mixture-of-experts (MoE) models

**Weaknesses:**

- The key weakness is explainly clearly where the benefit is coming from? I appreciate the author showing the Table 3 the ablation on removing several components, but its not clear which is the key component?
For example, main contributio comes from imitiation data or the dense feedback? Its not clear why training separately is an issue and additive helps?
- How good is the performance with first SFT and then RL with varying beta? Where is the comparison of KL which makes it unclear. Since,  whats the base policy being compared to?
- The reference policy in this case since SFT and RL is jointly trained needs to be specified and how far are you going from that for the alignment task.
- The authors show convergence analysis which is standard for non-convex objectives, but whats the additional term or component which tells us why it improves. So basically, just show what standard RL would do and what this approach would do which term is new and why?
- Why there are no much ablations on multiple datasets and also considering multiple other reward functions?

**Questions:**

See Weakness.

---

### Official Review · Reviewer_PcA2 · 2025-10-31

**Soundness:** 3
**Presentation:** 2
**Contribution:** 2
**Rating:** 4
**Confidence:** 3

**Summary:**

This paper introduces GRAO (Group Relative Alignment Optimization), a unified alignment framework that combines the benefits of Supervised Fine-Tuning (SFT) and Reinforcement Learning (RL) to achieve efficient and robust alignment for large language models (LLMs). The proposed optimization objective jointly leverages reference trajectories for imitation and self-sampled outputs for exploration by three key innovations, a multi-sample strategy, a new loss formulation, and reference-aware parameter updates. The evaluation shows the outperformance of the proposed framework across various alignment baselines, such as SFT, DPO, PPO and GRPO.

**Strengths:**

1. The methodology is illustrated clearly.
2. The convergence of the proposed optimization objective is proved theoretically.
3. The experiments cover various baselines and include more than one LLM family.

**Weaknesses:**

1. The hyperparameters $\beta$ and $\lambda$ are important. They balance exploration and imitation and control regularization strength. However, despite the default value, the author did not provide more information on these. Some ablation study is needed here for understanding how the algorithm synergizes SFT and RL.
2. The figures for the experimental results are not fine-grained. The vector format should be used in the final version of paper.
3. The theoretical bound in Equation 4 is not deeply discussed, and there is no corresponding experimental results to verify the message from this bound.
4. In Section 4.3,  the authors attribute Moonlight-16B MoE model's higher gain than the dense Qwen2.5-7B model to GRAO's ability to improve on sparse MoE architectures. I think this point is not supported very well by enough evidence. Maybe the results of more spare and dense models should be evaluated.

**Questions:**

1. For the optimal values of $\beta$ and $\lambda$, are they general or specific for the creation dataset and model?
2. For the metrics in the experiments, could the win rates applicable here? Why not use that?
3. Can you show the uncertainty of training dynamics in Figure 2 and 3?
4. Does the results from Figure 4b mean that the optimal value of $\beta$ and $\lambda$ should be adaptive?

---

### Official Review · Reviewer_df9z · 2025-11-01

**Soundness:** 2
**Presentation:** 1
**Contribution:** 2
**Rating:** 2
**Confidence:** 4

**Summary:**

This paper aims to study alignment capabilities within large language models by aligning a model to better produce aligned models, i.e., alignment for alignment. While the problem is timely, the paper’s core problem formulation, proposed method, and contributions are not clearly articulated. As a result, the novelty and significance are difficult to assess.

**Strengths:**

- Studying whether models can be trained to improve the alignment of other models is an important research question, particularly given the increasing emphasis on automated safety supervision and scalable oversight.

- The paper attempts to provide a mathematical formulation, which, if clarified and strengthened, could contribute to the emerging space of alignment studies.

**Weaknesses:**

- The framing (“alignment for alignment capability”) is vague and not presented in a way that builds intuition.

- It is unclear what concrete capability the paper is targeting or how success should be measured.

- The introduction does not present a crisp research question or conceptual insight.

- Contributions are not enumerated cleanly, and the storyline does not reveal what the reader should take away.

- Several equations do not follow standard conventions from related alignment and bilevel optimization literature. Example: In Equation (2), the left-hand side is expressed in terms of parameters $\theta$, but the right-hand side does not contain $\theta$,  this makes the expression ill-posed as an optimization problem.

-  It is unclear where performance gains originate from, and no ablations or mechanistic explanation are provided.

- The experiments do not convincingly isolate the contribution of the proposed method versus baseline tuning dynamics or training noise.

- There is no clear comparison with recent work on scalable oversight, teaching via RLHF, or alignment-via-distillation.

**Questions:**

Please refer to weaknesses.

---

### Note · Authors · 2025-11-29

I have read and agree with the venue's withdrawal policy on behalf of myself and my co-authors.